# Spatial Distribution Variation and Probabilistic Risk Assessment of Exposure to Fluoride in Ground Water Supplies: A Case Study in an Endemic Fluorosis Region of Northwest Iran

**DOI:** 10.3390/ijerph16040564

**Published:** 2019-02-15

**Authors:** Mahmood Yousefi, Farzaneh Baghal Asghari, Pietro Zuccarello, Gea Oliveri Conti, Aida Ejlali, Ali Akbar Mohammadi, Margherita Ferrante

**Affiliations:** 1Department of Environmental Health Engineering, School of Public Health, Tehran University of Medical Sciences, Tehran, Iran; mahmood_yousefi70@yahoo.com (M.Y.); asgharifarzane@gmail.com (F.B.A.); 2Department of Medical Sciences, Surgical and Advanced Technologies “G.F. Ingrassia”–Hygiene and Public Health, Laboratory of Environmental and Food Hygiene (LIAA), University of Catania, 95123 Catania, Italy; pietro.zuccarello@unict.it (P.Z.); marfer@unict.it (M.F.); 3Health Center of Urmia, Urmia University of Medical Sciences, Urmia, Iran; aidaejlali@yahoo.com; 4Department of Environmental Health Engineering, Neyshabur University of Medical Sciences, Neyshabur, Iran

**Keywords:** drinking water, fluoride, risk assessments, Hazard Quotient, spatial distribution

## Abstract

Prevalence of fluorosis is a worldwide public health issue, especially in the West Azerbaijan province of Iran. The aim of this study was to investigate fluoride concentration in drinking water resources within Maku city, in both the warm and cold seasons, to perform a health risk assessment. Fluoride were measured using UV-visible spectrophotometry. The spatial distribution was calculated by the software ArcGIS and Hazard Quotients (HQs) were calculated according to the US EPA method. The fluoride concentrations ranged between 0.29 to 6.68 and 0.1 to 11.4 mg/L in the cold and warm seasons, respectively. Based on this report, 30.64 and 48.15% of the samples revealed a fluoride level higher than the permissible level in the cold and warm seasons, respectively. Moreover, results showed that the HQ value in the warm season for different age groups was higher than the HQ value in the cold season. In both seasons, the non-carcinogenic risks of fluoride for the four exposed populations varied according to the order: children > teenagers > adults > infants. The HQ values for three age groups (children, teenager and adults) for both seasons were higher than 1 with a high risk of fluorosis. The results of this study, support the requests that government authorities better manage water supplies to improve health quality.

## 1. Introduction

Population growth, industrialization of societies, and increased droughts have led both a reduction of groundwater levels and increased pollution in drinking water sources. Among the pollutants of drinking water sources heavy metals, pesticides, nitrates, radon and fluoride can be mentioned [1,2,3,4,5]. Fluoride is an essential micronutrient for humans to grow bone and teeth at an early age. The human intake of fluoride occurs through water, soil and air but, among these, the main source of fluoride is drinking water [6,7,8]. Since fluoride is 100% water-soluble, the daily intake of fluoride depends on both concentration of fluoride in the drinking water and the daily intake of the same drinking water [7]. Fluoride intake is beneficial for human health only in a standard concentration range defined by the World Health Organization (1.5 mg/L). However, the WHO qualified this statement by recommending the limit be adapted to local conditions such as climate, water consumption and diet [9,10]. When fluoride is lower than the safe standards, it causes corrosion and tooth decay; on the other hand, higher concentrations of fluoride (>4 mg/L) in drinking water have been linked to physiological and pathological changes, such as dental and skeletal fluorosis [11]. Many studies has been done about the effects of fluoride on human health, including blood pressure, fertility, abortion, height and weight of babies, intelligence quotient and skeletal disorders [11,12,13,14,15,16,17]. Several water resources polluted by fluoride have been reported in China, South Africa, Mexico, Argentina and Iran (Khaf, Poldasht, Shout, etc.) [5,18,19,20,21]. Studies carried out in West Azerbaijan province [15], in northwest of Iran, show that the water contamination by fluoride is endemic due to the existence of basaltic lavas, at considerable depth, affecting the Sari Sou river with releases of fluoride in its waters [15]. Mohammadi et al. reported the prevalence of endemic fluorosis evidenced by the fact skeletal fluorosis of the people who live in areas with high fluoride concentration is 18.1% higher than in individuals who live in areas with low fluoride concentration in West Azerbaijan [11]. Karimzade et al. also carried out a study on the effect of the fluoride on the intelligence quotient (IQ) of children in two similar rural communities of Azerbaijan Province in Iran with drinking water with high and low fluoride levels and the results proved there is a significant linear trend for children in the high-drinking water F region to have a lower IQ [16]. It seems that the northern areas of Azerbaijan province can be considered as an endemic area for fluoride pollution. The high level of fluoride concentrations can be attributed to different minerals in the aquifer bed, and also the physicochemical properties of water. On the other hand, in many parts of the West Azerbaijan, high concentrations of fluoride have been observed frequently. This can be linked to the weathering and leaching of fluoride-containing minerals [11].

The most common risk management strategy for local authorities is to monitor the fluoride concentrations in the drinking water, as this is commonly assumed to be the predominant way of exposure. Health risk assessments (HRAs) have become a hot issue topic since the 1990s. HRAs consider a risk value as a numerical index for quantifying the magnitude and possibility of health hazards caused by harmful elements [6,22,23]. The simplest method is the calculation of a target Hazard Quotient (HQ), where the point estimate of the measured or predicted exposure concentration is divided by the threshold or safe level determined from a dose-response relationship [24]. A HQ less than or equal to 1 indicates that adverse effects are not likely to occur, and thus can be considered to have negligible hazard. HQs greater than 1 are not statistical probabilities of harm occurring. Due to specific geographical location of Maku city in West Azerbaijan province and, due to geological characteristics of its soils, the fluoride contamination and evaluation of fluoride exposure hazard for the non-carcinogenic diseases in the area need to be studied, so the aims of this research were:Identify the areas where high-fluoride waters developed through a geographic information system (GIS) and,Calculate according to the US EPA approach for the non-carcinogenic diseases risk assessment of residents exposed to fluoride through the ground water supplies.

## 2. Materials and Methods

### 2.1. Study Area and Sampling

The Maku area lies in the north of West Azerbaijan province, in the extreme northwest of Iran. The total study area is about 2302 km^2^, but only 650 km^2^ is covered by basaltic lava flows (Figure 1). The prevailing climate of the Maku area has semi-arid characteristics. Average annual precipitation at Maku station was 289 mm. Figure 1 shows all 65 sources of drinking water, including wells, springs and rivers that supply drinking water to 95 villages in the area. The study was carried out in both warm and cold season by March to May and by June to August (2014) of one year. The location of all sampling drinking water sources was recorded using a GPS device (GPSMAP 78s, Garmin, Olathe, KS, USA) and data were collected as latitude, longitude and angular distance.

The collected samples were taken from drinking water resources including wells and springs from 95 villages of the Maku city. A total of 130 samples were collected for two seasons in 2016. The water samples were collected in the sterile plastic containers 250 cc, which were washed twice before being sampled with the same water sampled and then transported to the laboratory. The samples were kept in refrigerator maintained at 4 °C. The sampling locations and their sources are given in Figure 1.

### 2.2. Determination of Fluoride in Samples of Drinking Water

Fluoride concentration of water samples was determined using the 8029 SPADNS method according to the instructions of the US EPA Standard Methods for the Examination of Water and Wastewater. The fluoride concentration was assessed using a spectrophotometer (DR-5000, Hach Company, Loveland, CO, USA) using an alternate wavelength in the 550–580 nm range.

The SPADNS method for fluoride determination involves the reaction of fluoride with a red zirconium-dye solution. The fluoride combines with part of the zirconium to form a colorless complex, thus bleaching the red color in an amount proportional to the fluoride concentration [23,24]. The method measures fluoride in the range between 0.0625–1.75 mg L^−1^ (*r* = 0.9993) and levels higher of this concentration were diluted and then measured. The obtained limits of determination (LOD) and quantification (LOQ) were 0.12 ppm and 0.37 ppm, respectively.

### 2.3. Calculation of Health Risk Assessment

A human health risk assessment is a method to calculate the nature and probability of adverse health effects in humans who may be exposed to chemicals in contaminated environmental media [3,25]. T quantitative health risk assessment of fluoride through consumption of drinking water was evaluated in the population of Maku city of West Azerbaijan province. Fluoride is classified, according to the IRCC classification, as a non-carcinogenic (D) group. The US Environmental Protection Agency (US EPA) provides a “Regional Screening Levels (RSLs) for Chemical Contaminants” online calculator [24,25]. For fluoride, we calculated the chronic risk for the resident population, expressed as Hazard Quotient (HQ—unitless), by consuming water. In particular, four age groups: infants (less than 2 years), children (2 to <6 years), teenagers (6 to <16 years) and adults (≥16 years) on the basis on physiological and behavioral differences according to the previous study of Yousefi et al. [24]. A HQ higher than 1 indicates the existence of a risk of developing non-carcinogenic effects during the lifetime, and it means that HQ is higher than the reference dose (RfD) defined as the maximum acceptable oral dose of a toxic substance.

Exposure to fluoride was calculated in these groups using Equation (1):(1)$$EDI=\frac{C_{f}\times C_{d}}{B_{w}}$$
where fluoride Estimated Daily Intake (EDI) is calculated based on the daily average consumption of drinking water (*C_d_*), concentration of F in drinking water (*C_f_*) and body weight (*B_w_*).

EDI is expressed in units of mg/kg BW/day. The water consumption data and body weight were estimated based on a validated questionnaire administered to the target groups (infants, children, teenager and adults). The average water consumption rates in infants (less than 2 years), children (2–6 years), teenagers (6–16 years) and adults (≥16 years) were 0.08, 0.85, 2 and 2.5 L day^−1^, respectively. Body weight of target groups were considered 10, 15, 50 and 78 kg, respectively (Table 1).

The non-carcinogenic risk of fluoride to human health can be expressed as hazard quotient (HQ) using Equation (2) [25]:(2)$$\mathrm{HQ}=\frac{EDI}{RfD}$$
where *RfD* indicates the reference dose. According to IRIS, the amount of *RfD* for fluorine (mg/kg/day) is 0.06 (USEPA, IRIS U), HQ indicates a risk level indicator for non-cancerous substances (HQ > 1 indicates adverse health effects are possible or dangerous conditions and HQ <1 indicating safe conditions for sensitive populations).

### 2.4. GIS

One of the most common methodologies for assessing variations in water geochemistry is geographic information system (GIS) which is a computer system for managing spatial data [26]. In this work the ArcGIS software (ESRI, Redlands, CA, USA) was used.

### 2.5. Statistical Analysis

All descriptive statistics such as average, standard deviation, minimum and maximum for the present study were calculated by using the Excel 2016 software (Microsoft, Redmond, WA, USA). Statistical analysis such as one-way ANOVA test analysis was done using IBM SPSS Statistics for Windows (Version 24.0, Release 2016, IBM Corp., Armonk, NY, USA). *p*-Values less than 0.05 considered statistically significant.

## 3. Results

### 3.1. Data Summary 

The data concerning the population and type of source of drinking water of villages in the study area is presented in Table 2.

### 3.2. Fluoride in Drinking Water Samples in Cold and Warm Seasons

In this study, 130 drinking water samples collected in two cold and warm seasons were analyzed for fluoride concentrations. The fluoride concentrations varied from 0.29 to 6.68 and 0.1 to 11.4 mg/L in the cold and warm seasons, respectively. Spatial distribution of fluoride in groundwater in the studied areas both in cold and warm seasons is reported in Figure 2. A comparison of our results expressed as mean fluoride concentrations in the studied area with the WHO guideline for fluoride in drinking water in the cold season is shown in Table 3. Based on this research, 30.64% of the samples had a fluoride level higher than the permissible level, 6.45% had less than the permissible limit and, 62.91% of the samples had a level within the optimum range of 0.5 to 1.5 ppm. In the warm season 48.15% of the samples had a fluoride level higher than the permissible level, 6.45% had less than the permissible limit, and 45.4% of the samples had a level within the optimum range of 0.5 to 1.5 ppm.

One factor analysis of sample paired *t*-test of data proved that concentration of fluoride in warm and cold seasons have a positive correlation during the periods in different village water samples (*p* < 0.05).

### 3.3. Exposure and Risk Assessment in Cold and Warm Seasons

The hazard quotient (HQ) and Estimated Daily intake (EDI) were calculated for four age groups in the different seasons, as shown in Table 4. The mean fluoride exposure of the villages’ residents showed high EDI levels during the warm season as compared to the cold season for the different age groups (infants, children, teenager and adults). Also, the HQ values in the warm season for the different age groups (infants, children, teenager and adults) were higher than the HQ values in the cold season.

For each group in this studied area in both seasons, the non-carcinogenic risks (HQ level) of fluoride for the four exposed populations varied in order: children > teenagers > adults > infants. Consequently, these groups of young people can be considered as a hypersensitive population.

The spatial distribution of HQ in the cold and warm seasons for the infant and children groups presented in Figure 3 shows that the HQ of fluoride in some regions, such as Takhteh Doz, Rand, and Alighandoo was higher than other regions such as Hajihasan and Hajo villages. Also we find from Table 3, that the fluoride non-carcinogenic-risk associated with the consumption of water in the two seasons for the infants groups were negligible because HQ was <1. The spatial distribution of HQ both in cold and warm seasons for teenagers and adult groups respectively is reported in Figure 4.

## 4. Discussion

In this study, 130 drinking water samples were analyzed for fluoride concentrations in two cold and warm seasons of 2016. The fluoride concentrations varied from 0.29 to 6.68 and 0.1 to 11.4 mg/L in the cold and warm seasons, respectively. A comparison of mean fluoride concentration in the studied area with respect to the WHO guideline for fluoride in drinking water in the cold season was shown in Table 2. Based on our research, 30.64% of the waters samples had a fluoride level higher than the permissible level, 6.45% had less than the permissible limit, and 62.91% of the samples had a level within the optimum limit of 0.5 to 1.5 ppm in the cold season, while in the warm season 48.15% of the samples had a fluoride level higher than the permissible level, only 6.45% had less than the permissible limit, and 45.4% of the samples had a level within the optimum limit of 0.5 to 1.5 ppm. (Table 2, Figure 2). In a similar investigation carried out by Keshsvarz [27], the measured fluoride concentrations varied from 1.29 to 3.1 and by 1.39 to 3.2 mg/L in spring and summer, respectively. This was confirmed also by our research. Ashghari et al., reported the average concentration of fluoride in warm and cold seasons in the ranges 0.01–3 and 0.01–4 mg/L, respectively [28].

In our study, the values of HQ and EDI for the studied groups including infants (0–2 years old), children (2–6 years old), teenagers (6–16 years old) and adults (≥16 years old) were calculated separately for each season in the study area (Table 3). Mean values of HQ for the studied groups, (infants, children, teenagers and adults) were 0.22, 1.56, 1.1, 0.88 (in the cold season) and 0.37, 2.6, 1.83 and 1.47, respectively, for the warm season (Table 3). In the cold season, the infant group had all HQ < 1 and in the children group, 58.06% had HQ values > 1. In the adolescents age group 25.8% of the samples had a HQ > 1. In the adults group 22.88% of the samples had HQ > 1. None of the samples of the four groups in the cold season had HQ > 1. In the warm season, HQ levels in the infant, children, teenagers and adult groups with HQ > 1 were 17.77%, 29/61%, 25.8% and 19.24%, respectively. In this season, in 11 samples, 17.74% of the samples all four age groups had HQ > 1. Therefore, the vulnerable residents were potentially exposed to non-carcinogenic risks. Considering that fluoride concentrations in drinking water are usually the main source of daily intake, so in this studied area fluoride exposure should be counted among the health problem concerns and special measures must be taken to decrease the fluoride concentration to minimize the adverse health effects on the residents. Results of the study of Keshavrz et al. about fluoride exposure and its health risk assessment in drinking water and primary food in the population of Dayyer (Iran) showed that HQ values of fluoride for the two age groups were 1.7 and 2 times greater than 1 in spring, and 1.9, and 2.4 in summer for adults and children, respectively [27]. Similar research was also reported by Chen et al. which showed infants as the most vulnerable group and reported that majority of samples of infants (72%) and children (60%) were exposed to the adverse impact of fluoride [21]. Fallahzadeh et al.’s study showed that 68.77% of the samples taken in Iran were within the standard range set by the WHO guidelines (0.5–1.5 mg/L). The mean (standard deviation) of fluoride in Ardakan, Azezar, Mehriz, Meybod, Taft and Yazd were 0.83 (0.31), 0.73 (0.41), 0.56 (0.20), 0.91 (0.32), 0.60 (0.32) and 0.64 (0.25), respectively. Among the studied counties, Ashkezar has the highest dispersion in terms of high concentration of fluoride. The hazard quotient (HQ) value for all age groups except children was less than 1, indicating a potential non-cancer risk of exposure to fluoride for this group [8]. The study conducted by Martinez showed that children from Zacatecas (Mexico) have a higher risk of non-carcinogenic health effects through consumption of drinking water with a high fluoride concentration [20]. Zhang et al. study showed that highest non-carcinogenic risk of fluoride exposure was again in the children group. All reported results were also confirmed by our study [29]. The results of Guissouma et al.’s study reported that young consumers in Tunisia, including infants and children, were more exposed to health risks caused by fluoride ingestion through the drinking water [22]. The HQ values of our study indicated that health risks assessment in relation to fluoride concentration, both for children and adults, are significant only from drinking water consumption and a potential risk of dental and skeletal fluorosis could be expected.

## 5. Conclusions

Contamination of drinking water by fluoride and its potential health implications remain a major public health issue worldwide, especially in the West Azerbaijan province of Iran. The results from a health risk assessment have demontrated quick decision-making tools to decrease environmental health problems. In this research, the fluoride concentration, EDI and HQ values have been calculated and the human risk have been estimated for fluoride in 95 drinking water resources in the rural area of Maku city in the cold and warm seasons. The fluoride concentrations varied from 0.29 to 6.68 and 0.1 to 11.4 mg/L in the cold and warm seasons, respectively. Based on this report, 30.64 and 48.15% of the samples had a fluoride level higher than the permissible level in the cold and warm seasons, respectively. Our results also showed the HQ value in the warm season for different age groups (infants, children, teenager and adults) was higher than the HQ value in the cold season. For each studied area in both seasons, the non-carcinogenic risks (HQ level) of fluoride for the four exposed populations varied in the order: children > teenagers > adults > infants. The hazard quotient (HQ) value for three age groups (children, teenagers and adults) for both seasons were higher than >1, a value that indicates a high risk of fluorosis due to excessive fluoride ingestion. The results of this study also support the request that government authorities better manage water supplies with the aim to provide water with low fluoride concentration.

## Figures and Tables

**Figure 1 ijerph-16-00564-f001:**
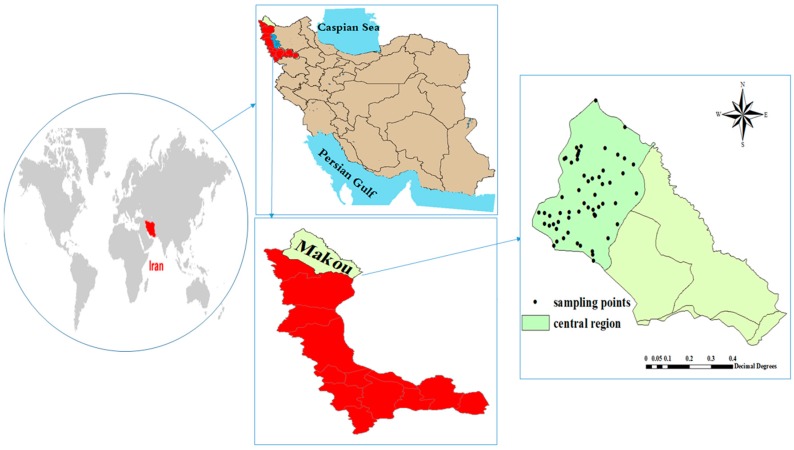
Study area and sampling location.

**Figure 2 ijerph-16-00564-f002:**
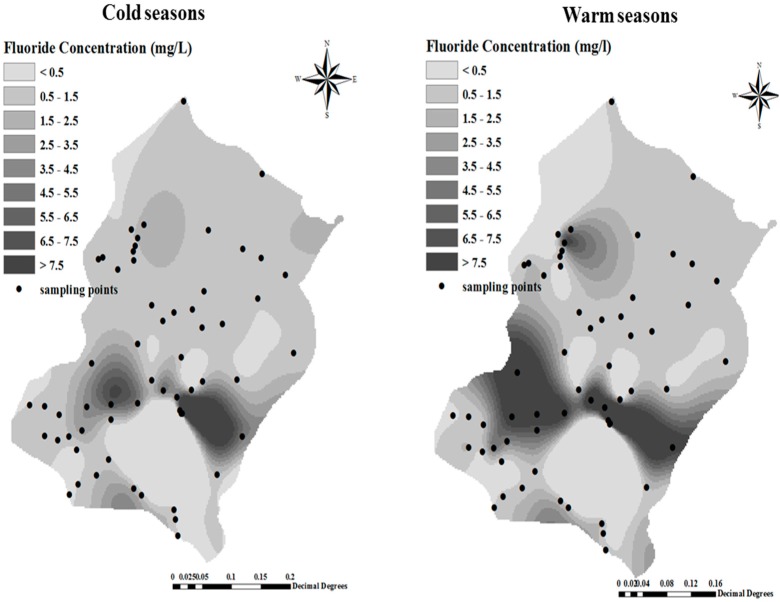
Spatial distribution of fluoride in groundwater in the studied areas in cold and warm seasons.

**Figure 3 ijerph-16-00564-f003:**
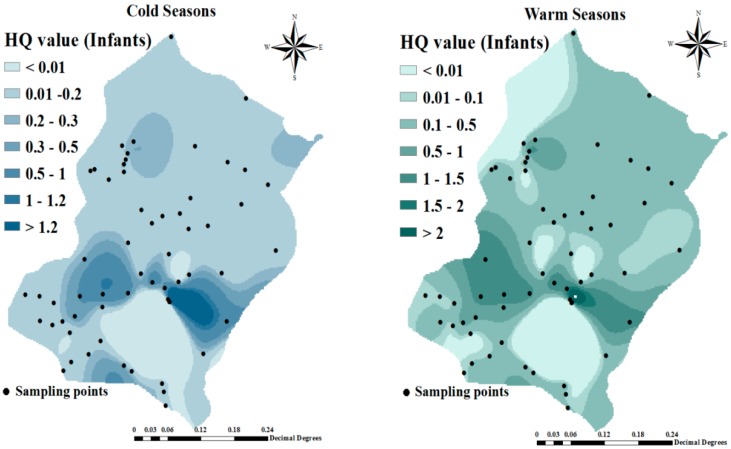
Spatial distribution of hazard quotient (HQ) in cold and warm seasons for infant and children groups.

**Figure 4 ijerph-16-00564-f004:**
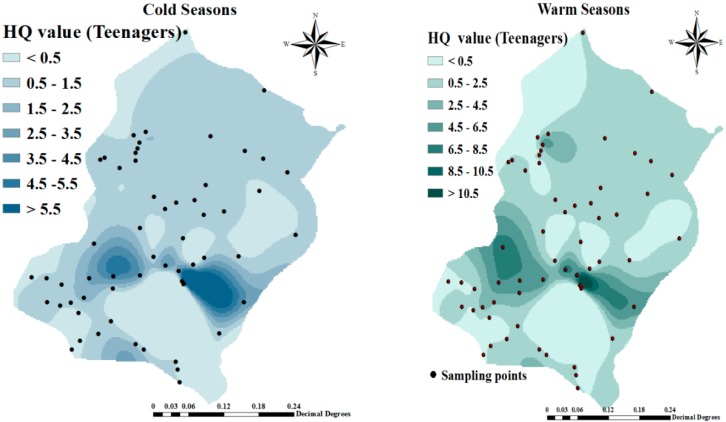
Spatial distribution of HQ in cold and warm seasons for teenagers and adult groups.

**Table 1 ijerph-16-00564-t001:** Parameters used in the present study for Hazard Quotient (HQ) calculations.

Risk Exposure Factors	Values for Groups				Unit
	Infants	Children	Teenagers	Adults	
C_f_					mg/L
C_d_	0.08	0.85	2	2.5	Liter/day
B_w_	10	15	50	78	Kg
RfD	0.06	0.06	0.06	0.06	mg/kg.day

C_f_: concentration of F in drinking water; C_d_: daily average consumption of drinking water; B_w_: body weight; RfD: reference dose.

**Table 2 ijerph-16-00564-t002:** Summary of the population and type of source drinking water of villages in the study area (Maku).

Sampling Location	Source	Population	Sampling Location	Source	Population	Sampling Location	Source	Population
Ghezel Dagh Kord	Deep well	927	Ghara Khach	Spring	281	Haji Hassan	Spring	-
Hassain Bozorgh	Spring	203	Turkan	Spring	-	-	Spring	-
Hassain Kochekh	Manual well	207	Dibak	Spring	128	Ghrik	Spring	273
Mirza Khalil	Spring	171	Baroon	Spring	174	Yarim gayeh olya	Deep well	278
Tazakand adaghan	Manual well	147	Ghoosh	Spring	423	Kharman Yeri	Spring	491
Hasso shaki	Manual well	173	Kholkhola	Spring	-	Ghori shakak	Manual well	699
Hesar	Spring	392	Aghgol	Spring	1067	Kishmish tapa	Deep well	3623
Isa khan	Manual well	170	Gomshor aghgol	Spring	194	Chamanloo	Spring	29
Ghala zagasi	Semi-deep well	-	Yikhilgan	Spring	195	Aghbilagh Chamanlo	Spring	118
Tika kord	Semi-deep well	313	Molik	Spring	-	Tika ajam	Well	117
Adaghan	Semi-deep well	-	Ghara Bilagh	Spring	110	Jan aziz	Spring	-
Torkma	Spring	282	Markmi	Spring	146	Mail kandi	Spring	7
Sangar	Deep well	651	Bash Kand	Spring	485	Ghala jogh	Spring	-
Takhteh Doz	Well	328	Ghojat	Well		Para khodak	Spring	-
Hasso shiri	Spring	106	Tlim Khan	Manual well	145	Kosa kandi	Spring	378
Danoye Bozorgh	Spring	451	Ghishlagh	Spring	111	Mohammad kandi	Spring	-
Kahriz ghalasi	Spring	-	Shorick	Spring	-	Baqcheh Jooq	Well	-
Alighandoo	Spring	175	Hajoo	Spring	-	Gamos	Spring	-
Ali abad	Spring	205	Goal ali	Spring	209	Rand	Spring	340
Mohammad abad	Well	92	Mus	Spring	132	Rand	Well	340
Jaganloyeh ajam	Spring	7	Tajdoo	Spring	134			

**Table 3 ijerph-16-00564-t003:** The fluoride concentration (mg/L) in drinking water sources of the studied area, expressed as means ± standard deviation, min and max.

Statistical Analysis	Fluoride Concentration	
	Cold Season	Warm Season
Number of samples	62.00	62.00
Max	6.68	11.14
Min	0.29	0.1
Mean	1.65	2.75
SD	1.44	3.33
WHO Guideline	1.5	
Maximum allowable	1.5	
Minimum allowable	0.5	
Percentage of fluoride concentration low 0.5 mg/L	6.45	6.45
Percentage of fluoride concentration above 1.5 mg/L	25.80	25.80
Percentage of fluoride concentration above 5 mg/L	4.84	19.35

**Table 4 ijerph-16-00564-t004:** HQ and Estimated Daily intake (EDI) values in cold and warm seasons for different for different age groups (infants, children, teenager and adults).

Different Age Groups	HQ		EDI	
	Cold	Warm	Cold	Warm
Mean infants group	0.220	0.366	0.013	0.022
Mean children group	1.558	2.596	0.093	0.156
Mean teenagers group	1.100	1.832	0.066	0.110
Mean adults group	0.881	1.468	0.053	0.088
Minimum infants group	0.038	0.013	0.002	0.001
Minimum children group	0.271	0.094	0.016	0.006
Minimum teenagers group	0.191	0.067	0.011	0.004
Minimum adults group	0.153	0.053	0.009	0.003
Maximum infants group	0.891	1.485	0.053	0.089
Maximum children group	6.312	10.520	0.379	0.631
Maximum teenagers group	4.455	7.426	0.267	0.446
Maximum adults group	3.570	5.950	0.214	0.357
Percentage(HQ > 1) in the group infant	0	17.74	-	-
Percentage(HQ > 1) in the group children	58.06	61.29	-	-
Percentage(HQ > 1) in the group teenagers	25.8	25.8	-	-
Percentage(HQ > 1) in the group adults	17.74	24.19	-	-
Percentage of samples	0	17.74	-	-
in the four age groups of HQ > 1				

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
