# Peer review of "Spatial Distribution Variation and Probabilistic Risk Assessment of Exposure to Fluoride in Ground Water Supplies: A Case Study in an Endemic Fluorosis Region of Northwest Iran"

_ijerph, 2019, doi:10.3390/ijerph16040564_

Round 1
Reviewer 1 Report
the topic is of great international interest but there are some significant comments needed particularly linking the environmental conditions to health risk.
introduction:
I think the comment about 100% solubility of fluoride is misleading LN41. many minerals contain fluoride and its solubility is relatively low in groundwater systems. it is high in the region due to the high presence of percolating fluids and the nature of the rock in contact. there needs to be an overview of fluoride controls in groundwater and a specific review of the conditions causing the problem in the location being studied.
the background also needs better discussion of current knowledge on fluorosis in the region - motivation for the study is rather weak. also given the very dispersed population (table 1) the public health drivers could be explained.
Materials and methods
study location
details of the hydrogeology, given the different sources of water - spring v wells and habitation - many dispersed villages. population details - table 1 suggests very small groups and samples collected repeatedly. why do we not have e.g. age profile etc information - this would inform the HQ analysis later
determination of F
quality assurance data needed reproducibility and standard reference materials. what other water quality parameters were measured? EC and pH would be a minimum and help in the interpretation.
water sampling - a single set of amples collected for each season. the reproducibility should have been assessed with multiple samples at a number of locations.
health risk determination
this follows basic USEPA formula. there needs to be a more local assessment given the assumptions of consumption and weight. these could be relatively easily assessed during sampling - revision /explanation of the relevance of these values in the study area
GIS - the use of GIS is not clear. the maps produced show interpolated distribution of F this cannot be true as the water sources are variable and the hydrogeology of the region is unlikely to come from one aquifer. so using spot samples from wells and springs and then interpolating it is totally in-appropriate to display the data. also the only water chemistry provided id "F" content. this is almost meaningless without other parameters so the "geochemistry" is not clearly described
statistical analysis
descriptive statistics are mentioned but nothing about other comparisons. the difference between warm and cold is an important observation - this should be backed up/justified using a suitable test method (simple paired t-test?)
results
these need to be revised given methodology comments above.
more information about population structure, F levels need backing up from a replication perspective, hazard quotient calculation needs revision/explanation in the context of the population being studied. i would anticipate that there is information to help to make this assessment more relevant.
spatial distribution of HQ is meaningless using these map displays and interpolation .there are better ways to display this information - subject to revisions above
generally - i would encourage the group to engage with regional geological expertise to bring better understanding to the water conditions
Author Response
Dear reviewers thank you to have highlighted the lacks on behalf of all coauthors. We revised the manuscript according the provided suggestions and comments. I thank you for the chance!
Comments of Reviewer 2 . Dear Reviewer, I thank you for your suggestions. The paper was revised according to your suggestions and comments.
· I think the comment about 100% solubility of fluoride is misleading LN41. Many minerals contain fluoride and its solubility is relatively low in groundwater systems. It is high in the region due to the high presence of percolating fluids and the nature of the rock in contact. RE: We revised the text and more informations about your comments are provided in introductions and material and method sections.
· There needs to be an overview of fluoride controls in groundwater and a specific review of the conditions causing the problem in the location being studied. RE: I thank you the reviewers to highlight this lacks. DONE. We revised the text in introduction section.
· The background also needs better discussion of current knowledge on fluorosis in the region - motivation for the study is rather weak. RE: DONE. We add modified in the text introduction.
· Also given the very dispersed population (table 1) the public health drivers could be explained. RE: considering that the sample water collected from two category village including central village and sub central village. Also in some villages the population has been minimized due to migration to cities for searching work.
Materials and methods
· study location: details of the hydrogeology, given the different sources of water - spring v wells and habitation - many dispersed villages. population details - table 1 suggests very small groups and samples collected repeatedly. why do we not have e.g. age profile etc information - this would inform the HQ analysis later. RE: we don’t have age profile and other information.
· determination of F: quality assurance data needed reproducibility and standard reference materials. what other water quality parameters were measured? EC and pH would be a minimum and help in the interpretation. RE: Dear Reviewer, our target in study was monitoring of fluoride concentrations in drinking water samples and calculate risk assessment in the region, therefore other parameter (such as pH,EC,..) weren’t monitored
· water sampling - a single set of samples collected for each season. The reproducibility should have been assessed with multiple samples at a number of locations. RE: we can add this lack as weakness in discussions.
health risk determination
· this follows basic USEPA formula. there needs to be a more local assessment given the assumptions of consumption and weight. these could be relatively easily assessed during sampling - revision /explanation of the relevance of these values in the study area. Re: we modified table 1, in the text were reported all data and informations about assumptions of consumption and weight of populations.
· GIS - the use of GIS is not clear. the maps produced show interpolated distribution of F this cannot be true as the water sources are variable and the hydrogeology of the region is unlikely to come from one aquifer. So using spot samples from wells and springs and then interpolating it is totally in-appropriate to display the data. also the only water chemistry provided id "F" content. this is almost meaningless without other parameters so the "geochemistry" is not clearly described. RE: Dear Reviewer our target in study was monitoring of fluoride concentrations in drinking water samples and calculate risk assessments in the region, therefore other parameter (such as pH,EC,..) weren’t monitored.
Statistical analysis
· Descriptive statistics are mentioned but nothing about other comparisons. The difference between warm and cold is an important observation - this should be backed up/justified using a suitable test method (simple paired t-test?). RE : it done in the material method and result section
Results
These need to be revised given methodology comments above. RE: DONE.
More information about population structure, F levels need backing up from a replication perspective, hazard quotient calculation needs revision/explanation in the context of the population being studied. I would anticipate that there is information to help to make this assessment more relevant. RE: DONE. We added more informations about population structure in modified Table 2
Spatial distribution of HQ is meaningless using these map displays and interpolation .there are better ways to display this information - subject to revisions above. RE: Dear reviewer, we using these figures because of interest by manager of public health. The map is more easy to used better compared to table, for the workers of public health these pictures are more concise and useful.
· Generally - I would encourage the group to engage with regional geological expertise to bring better understanding to the water conditions. Dear Reviewer, our target in study was monitoring of fluoride concentration in drinking water samples and calculate risk assessment in the region, therefore other parameter (such as pH,EC,..) were not monitoring, and currently we haven’t funding to engage a geological expertise.
Reviewer 2 Report
This is an interesting article on the measurement of fluoride intake in various waterways in Iran. It is clearly a very important, and useful topic in understanding fluoride levels and their effect on health in different parts of the population.
However, there are a number of improvements to your article that could raise it from being okay, through to being really great.
The first is the use of the Hazard Quotient method to highlight risk. Unfortunatley, the HQ method cannot show level of risk, as risk (when measured, and according to international standards) is based on probability and consequence. This isn't the case when using the HQ method (which is based on estimated daily intake and the reference dose. A great article on linking HQ to risk is here: https://doi.org/10.1016/S0261-2194(00)00086-7 (Solomon, Giesy, Jones "Probabilistic risk assessment of agrochemicals in the environment).
I would consider removing references to risk unless you are actually measuring it. A section on why you use the Hazard Quotient method, as well as the strengths and weaknesses of the method would be useful. Also, you should clearly provide more information on how 'cold' and 'warm' seasons were defined (ie. between what dates?)
Line, and comments:
19: consider removing first sentence; it's obvious.
20-22: poor grammar; consider revising
23: First use of HQ should be expanded
27: cap. error ("Season")
27-29: rewrite, unclear what you are trying to say here
55: "your waters" - incorrect usage
71: replace 'were' with 'was'
91: 'water' written twice
97: indent in the middle of paragraph
Figure 1: reference for the figure, or is this your own?
139: "makes the word easier" - too vague. Need to be more specific.
143: consider changing "estimated" to "calculated"
160: "showed" should be "shown"
Table 3: could be shown more clearly. Minimum adults vs. Maximum adults group- unclear what this means.
Poor grammar throughout document, consider reviewing.
Overall, interesting and important piece, but could do with more refining.
Author Response
Dear reviewers thank you to have highlighted the lacks on behalf of all coauthors. We revised the manuscript according the provided suggestions and comments. I thank you for the chance!
Comments for Reviewer 2
Dear Reviewer, I thank you for your suggestions. The paper was revised according to your points.
This is an interesting article on the measurement of fluoride intake in various waterways in Iran. It is clearly a very important, and useful topic in understanding fluoride levels and their effect on health in different parts of the population. However, there are a number of improvements to your article that could raise it from being okay, through to being really great.
- The first is the use of the Hazard Quotient method to highlight risk. Unfortunatley, the HQ method cannot show level of risk, as risk (when measured, and according to international standards) is based on probability and consequence. This isn't the case when using the HQ method (which is based on estimated daily intake and the reference dose. A great article on linking HQ to risk is here: https://doi.org/10.1016/S0261-2194(00)00086-7 (Solomon, Giesy, Jones "Probabilistic risk assessment of agrochemicals in the environment).
- I would consider removing references to risk unless you are actually measuring it. A section on why you use the Hazard Quotient method, as well as the strengths and weaknesses of the method would be useful. RE: Done. We adjusted all definitions to avoid misunderstandings, and we provide more informations about HQ both in introduction and in materials and methods sections.
- Also, you should clearly provide more information on how 'cold' and 'warm' seasons were defined (ie. between what dates?): Dear Reviewer, I thank you for your suggestions. The paper was revised according to your points improving the result section
- Line, and comments:
- 19: consider removing first sentence; it's obvious. RE: DONE ,
- 20-22: poor grammar; consider revising RE: DONE
- 23: First use of HQ should be expanded RE: DONE
- 27: cap. error ("Season") RE: DONE
- 27-29: rewrite, unclear what you are trying to say here RE: DONE
- 55: "your waters" - incorrect usage RE: It is corrected. DONE
- 71: replace 'were' with 'was' RE: It is corrected. DONE
- 91: 'water' written twice RE: DONE
- 97: indent in the middle of paragraph RE: DONE
- Figure 1: reference for the figure, or is this your own? RE: All figures are original because self made.
- 139: "makes the word easier" - too vague. Need to be more specific. RE: DONE
- 143: consider changing "estimated" to "calculated " RE: DONE
- 160: "showed" should be "shown" RE: DONE
- Table 3: could be shown more clearly. Minimum adults vs. Maximum adults group- unclear what this means. RE: DONE, we adjusted the title of table
Round 2
Reviewer 1 Report
i am happy with the response and modifications incorporated in the revised version
Reviewer 2 Report
Much better! Your paper is reading much more coherently, and well put together. Thank you for taking the time to consider the review comments.
The only outstanding element is the english writing. There are still numerous english-language and formatting mistakes throughout the document that would just need to be fixed before publication.